# Ultra-Compact Low-Pass Spoof Surface Plasmon Polariton Filter Based on Interdigital Structure

**DOI:** 10.3390/mi14091687

**Published:** 2023-08-29

**Authors:** Zhou-Hao Gao, Xin-Shuo Li, Man Mao, Chen Sun, Feng-Xue Liu, Le Zhang, Lei Zhao

**Affiliations:** 1JSNU SPBPU Institute of Engineering, Jiangsu Normal University, Xuzhou 221116, China; 3020202814@jsnu.edu.cn (Z.-H.G.); 3020211337@jsnu.edu.cn (X.-S.L.); 2School of Physics and Electronic Engineering, Jiangsu Normal University, Xuzhou 221116, China; 3020215317@jsnu.edu.cn (M.M.); zhangle@jsnu.edu.cn (L.Z.); 3School of Computer Science and Technology, Jiangsu Normal University, Xuzhou 221116, China; 3020200309@jsnu.edu.cn; 4Jiangsu Xiyi Advanced Materials Research Institute of Industrial Technology, Xuzhou 221400, China; 5School of Information and Control Engineering, China University of Mining and Technology, Xuzhou 221116, China; leizhao@cumt.edu.cn

**Keywords:** ultra-compact filter, low-pass filter, spoof surface plasmon polariton, interdigital structure

## Abstract

An ultra-compact low-pass spoof surface plasmon polariton (SSPP) filter based on an interdigital structure (IS) is designed. Simulated dispersion curves show that adding the interdigital structure in an SSPP unit effectively reduces its asymptotic frequency compared with traditional and T-shaped SSPP geometries, and the unit dimensions can be conversely reduced. Based on that, three IS-based SSPP units are, respectively, designed with different maximum intrinsic frequencies and similar asymptotic frequencies to constitute the matching and waveguide sections of the proposed filter, and the unit number in the waveguide section is adjusted to improve the out-of-band suppression. Simulation results illustrate the efficient transmission in the 0~5.66 GHz passband, excellent out-of-band suppression (over 24 dB) in the 5.95~12 GHz stopband and ultra-shape roll-off at 5.74 GHz of the proposed filter. Measurement results on a fabricated prototype validate the design, with a measured cut-off frequency of 5.53 GHz and an ultra-compact geometry of 0.5 × 0.16 *λ*_0_^2^.

## 1. Introduction

Surface plasmon polaritons (SPPs) can be excited from a metal medium by photons in the optical bands, with efficient propagation along the surface of the metal medium and rapid attenuation in the normal direction, and can be therefore applied in optical devices [1,2,3,4,5]. In the terahertz and microwave bands, however, the metal medium acts as a perfect electrical conductor (PEC) instead of plasma with a negative dielectric constant, and the SPP cannot be directly excited [6]. A spoof surface plasmon polariton (SSPP) can be excited from a metal plate with periodic holes, slots, rings or varactor diodes in the terahertz and microwave bands, and its propagation and attenuation characteristics are similar to those of an SPP [7,8,9,10,11,12]. Therefore, the SSPP structure has been applied in several microwave/terahertz devices including antennas [13,14,15,16], waveguides [17,18,19], power splitters/combiners [20,21,22] and various filters [23,24,25,26,27,28,29,30].

Among the existing applications of SSPP structures, the planar band-pass and low-pass SSPP filters operating in the microwave bands are hot research topics. However, for those filters with original SSPP structures, the filter sizes can be relatively large. Reference [24] proposed a low-pass SSPP filter consisting of double-sided corrugated Greek-cross fractal units with a cut-off frequency of 4.6 GHz and a filter size of 3.18 × 0.38 *λ*_0_^2^, where *λ*_0_ represents the wavelength in vacuum at the cut-off frequency. In reference [25], a defected ground was employed in a low-pass SSPP waveguide to decrease its cut-off frequency from 10.4 GHz to 8.9 GHz, but its size (3.12 × 0.34 *λ*_0_^2^) was still large. In reference [26], the half-mode substrate integrated waveguide structure was employed to halve the width of a band-pass 2.5-D SSPP transmission line, and the obtained geometrical dimension was still 1.87 × 0.47 *λ*_0_^2^. Reference [23] introduced a band-pass SSPP filter working in 0.25~4.5 GHz, and its meander-line structure reduced its geometric size to 1.31 × 0.19 *λ*_0_^2^. Reference [29] presented a low-pass filter using double-layered SSPPs with a cut-off frequency of 4.84 GHz, and the employed interdigital strips led to an ultra-compact geometry of 1.51 × 0.21 *λ*_0_^2^. Therefore, using an interdigital structure (IS) or a meander-line structure can effectively decrease the size of the band-pass or low-pass SSPP filters.

This paper presents a novel ultra-compact low-pass SSPP filter based on an IS. Firstly, the geometry of an IS-based SSPP unit is designed and compared with traditional and T-shaped SSPP units in terms of asymptotic frequency through simulations in CST Microwave Studio, and the effects of the dimensional parameters on the asymptotic frequency are investigated and analyzed based on the simulated dispersion curves. Secondly, three types of units are designed for the matching/waveguide sections, and the unit number in the waveguide section is optimized through simulations. Lastly, a prototype filter is fabricated and measured to validate the design. The proposed filter has excellent low-pass characteristics and an advantageous ultra-compact planar geometry, and can therefore be integrated in large-scale circuits for new-generation wireless communication networks.

## 2. Unit Design

The geometry of a traditional SSPP unit is shown in Figure 1a. The substrate material was selected to be FR-4 (dielectric constant *ε_r_* = 4.3, loss tangent tan *δ* = 0.025 and thickness *t_s_* = 0.508 mm), and both the patch and ground were copper layers with a thickness of 0.018 mm. Based on simulation in CST Microwave Studio, its dimensional parameters were determined as *p* = 2.5 mm, *h* = 2 mm, *d* = 0.1 mm, *w* = 0.2 mm. Its simulated dispersion curve is shown in Figure 2 with an asymptotic frequency of 16.7 GHz.

Inspired by reference [29], a T-shaped SSPP unit was used for comparison, as shown in Figure 1b. It was designed with the same materials and values of parameters *p*, *h* and *w* of the traditional unit, and parameters *g*_1_ and *w*_1_ were, respectively, optimized to be 0.2 and 0.1 mm. As is well known, the cut-off frequency of a two-port *LC* network is basically inversely proportional to $\sqrt{LC}$ where *L* and *C* are, respectively, the equivalent inductance and capacitance. The capacitance between the two L-shaped arms in the T-shaped SSPP unit is clearly larger than that between the two straight arms in the traditional SSPP unit. The increased capacitance explains the reduced asymptotic frequency of the T-shaped SSPP unit (11.4 GHz), as shown in the simulated dispersion curve in Figure 2.

Based on these conclusions and the verified relationship between equivalent capacitance and cut-off frequency, an interdigital structure was employed to further decrease asymptotic frequency. The geometry of the proposed IS-based SSPP unit is shown in Figure 1c. It keeps the same materials and values of parameters *p*, *h* and *w* of the traditional and T-shaped units for comparison, and parameters *g*_2_ and *w*_2_ were optimized through CST simulations to be 0.2 and 0.1 mm, respectively. The simulated dispersion curve of the IS-based SSPP unit is shown in Figure 2 with an asymptotic frequency of 6 GHz. Compared with the traditional and T-shaped SSPP units, the IS-based SSPP unit obviously has a much larger equivalent capacitance because of its interdigital structure, and therefore has a lower asymptotic frequency. Conversely, the size of the IS-based SSPP unit is much smaller than those of its traditional and T-shaped counterparts given the same asymptotic frequency.

Figure 3 shows the simulated curves of the asymptotic frequency with respect to parameters *p* and *g*_2_ for the IS-based SSPP unit. When *p* increases, the increased length of the interdigital parts of its two arms leads to increased equivalent capacitance, and asymptotic frequency is therefore decreased. When *g*_2_ increases, however, the increased gaps between the horizontal strips of its two interdigital arms leads to a decrease in the equivalent capacitance of the unit, and the asymptotic frequency is accordingly increased. Ideally, the cut-off frequency for the IS-based SSPP unit can be further reduced by increasing *p* or decreasing *g*_2_. However, a lower *g*_2_ requires higher fabrication accuracy with higher production cost, and increasing *p* can lead to an increase in the filter length, which is obviously not wanted in this work. Therefore, their values need to be chosen considering the balance between cut-off frequency, practical fabrication accuracy and overall length of the designed filter.

## 3. Geometry Design of Proposed Filter

Based on the designed IS-based SSPP unit, the geometry of an ultra-compact low-pass SSPP filter is shown in Figure 4. In Regions I and V, the microstrip lines act as the input and output ports, and the corresponding dimensional parameters are optimized to match the 50 Ω impedance of the cable as *w_s_* = 1 mm, *l*_1_ = 1 mm, *l*_2_ = 2 mm. In Regions II and IV, two types of IS-based SSPP units (denoted as U1 and U2) are designed based on the parameter studies in the last section to act as the matching section. In Region III, five consecutive identical SSPP units (denoted as U3s) constitute the waveguide section. The values of the dimensional parameters of U1, U2 and U3 are optimized through CST simulations and listed in Table 1.

Figure 5 shows the simulated dispersion curves of U1, U2 and U3. The dimensional parameters of U3 keep the same values obtained in Section 2 with a maximum intrinsic frequency of 6.03 GHz. For U1 and U2, parameters *p* and g_2_ are adjusted for matching between the microstrip line and the U3s, and their maximum intrinsic frequencies are, respectively, tuned to 6.86 and 6.23 GHz. The observed negative group velocities for all three units can be explained by the strong coupling effect and the large capacitance and inductance inside the IS-based unit geometries, as reported in reference [27]. Additionally, all three units are tuned to be of similar asymptotic frequencies around 6 GHz to guarantee the ultra-sharp roll-off of the proposed filter.

Figure 6 shows the simulated |S_11_| and |S_21_| curves of the insertion and return losses with different unit numbers N in Region III. It can be observed that changing N does not directly lead to changes in the cut-off frequency. In the passband, the insertion loss curve basically remains unchanged when N increases from 3 to 5, and the peak of the return loss curve is also largely unaffected. In the high-frequency stopband, no significant change is observed on the return loss curve when N increases, but the peak of the insertion loss curve drops. On the other hand, increasing N directly leads to an increase in the filter length. Therefore, N is determined to be 5 to achieve both a decent out-of-band suppression over 24 dB and a relatively small filter length.

## 4. Simulation Results of Proposed Filter

The simulated curves of the insertion and return losses of the proposed filter are shown in Figure 7. In the 0~5.66 GHz passband, the fact that the simulated insertion and return losses are, respectively, below 1 dB and above 14 dB proves high transmission efficiency. Also, the simulated cut-off frequency (5.74 GHz) is slightly lower than the simulated asymptotic frequencies of U1, U2 and U3 because of the capacitive coupling between adjacent units in the filter. In the 5.95~12 GHz stopband, the simulated |S_21_| basically stays below −24 dB, and thus indicates excellent out-of-band suppression. Additionally, the observed minor degradation at 11.8 GHz can be explained by the resonance of a high-order mode of the IS-based SSPP units.

On the other hand, the roll-off rate F of the filter can be defined as below to evaluate its roll-off characteristic [31]:(1)$$F=\left|\frac{\left(-3\;\mathrm{dB}\right)-\left(-25\;\mathrm{dB}\right)}{f_{L}-f_{H}}\right|$$
where *f_L_* and *f_H_* are the frequencies where |S_21_|, respectively, equals −3 and −25 dB. The calculated *F* based on simulation data is 105 dB/GHz, and therefore proves the ultra-sharp roll-off of the proposed filter.

Figure 8 shows the simulated E-field and surface current distributions of the proposed filter at 5.58 and 5.99 GHz. The input and output ports are Port 1 and Port 2, respectively. The investigated plane for the E-field keeps a 1 mm distance above the patch surface. At 5.58 GHz, which is below the cut-off frequency, it can be observed that the SSPP mode is excited and transmitted through all regions. However, at 5.99 GHz, which is above the cut-off frequency, the SSPP mode basically only exists in Region II and the first two units of Region III, and it can barely propagate to Port 2. Therefore, the low-pass characteristics of the proposed filter are verified.

## 5. Prototype Fabrication and Measurements of Proposed Filter

A prototype of the proposed filter was fabricated for measurement verification. The photos of the fabricated prototype are shown in Figure 9. In practical measurements, two SMA connectors were soldered at the two ports of the prototype.

The curves of the insertion and return losses of the prototype were measured by a vector network analyzer (Keysight N5234B) and are shown in Figure 10. The measured curves of the proposed filter show a reasonable agreement with simulations, and therefore prove its high transmission efficiency in the low-frequency passband and high out-of-band suppression in the high-frequency stopband. The measured cut-off frequency of the proposed filter is 5.53 GHz, which is lower than the simulated result. The minor shift between the measured and simulated cut-off frequencies can be attributed to the limited fabrication accuracy.

## 6. Conclusions

An ultra-compact low-pass SSPP filter based on an interdigital structure was introduced. Simulation results reveal that the designed IS-based SSPP units can effectively decrease the maximum intrinsic frequency, and the designed filter is therefore of an ultra-compact geometry with efficient transmission in the passband, excellent out-of-band suppression and ultra-sharp roll-off. Simulations show that *p*, *g*_2_ and N are the determining parameters for the proposed filter, and their values are optimized with a full consideration of the balance between filter performance, fabrication accuracy and filter length. The comparison between the proposed filter and other band-pass and low-pass SSPP filters operating in the microwave bands is shown in Table 2. The insertion loss in the passband and the out-of-band suppression of the proposed filter are basically on the same level as those of other competitors. However, the proposed filter is shorter and narrower, which are very advantageous for its integration in radio-frequency circuits.

## Figures and Tables

**Figure 1 micromachines-14-01687-f001:**
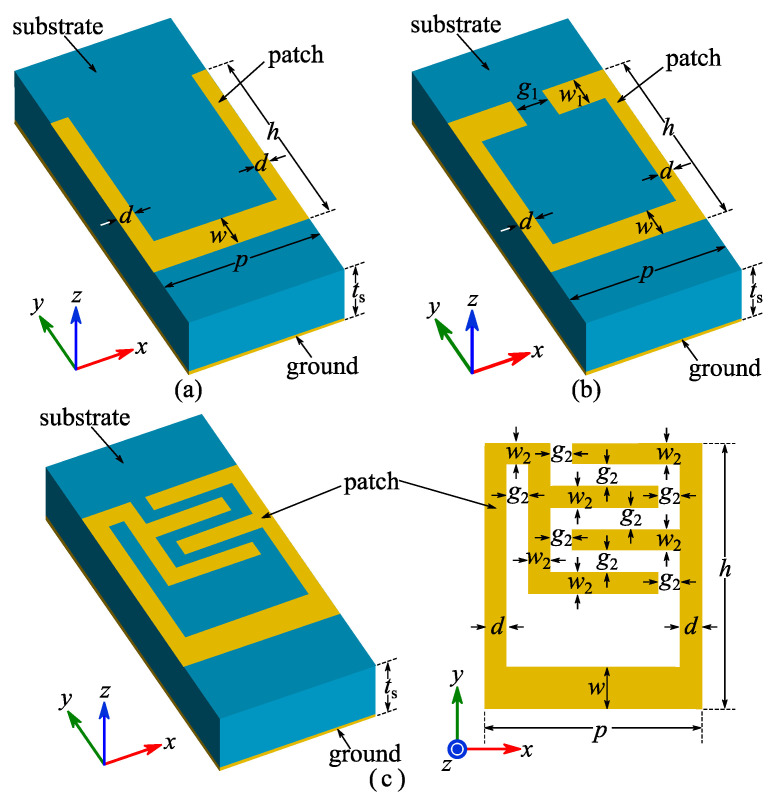
Geometries of (**a**) traditional, (**b**) T-shaped and (**c**) IS-based SSPP units.

**Figure 2 micromachines-14-01687-f002:**
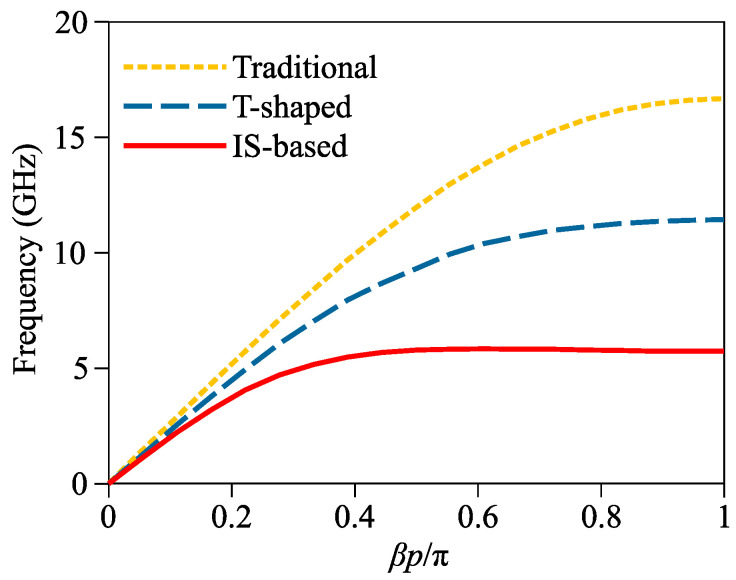
Simulated dispersion curves of traditional, T-shaped and IS-based SSPP units.

**Figure 3 micromachines-14-01687-f003:**
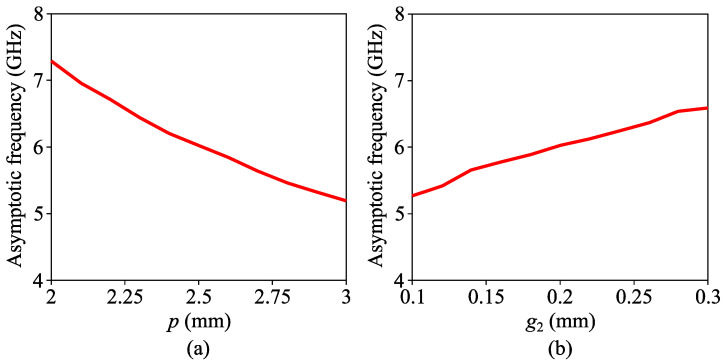
Simulated curves of asymptotic frequency with respect to (**a**) *p* and (**b**) *g*_2_.

**Figure 4 micromachines-14-01687-f004:**
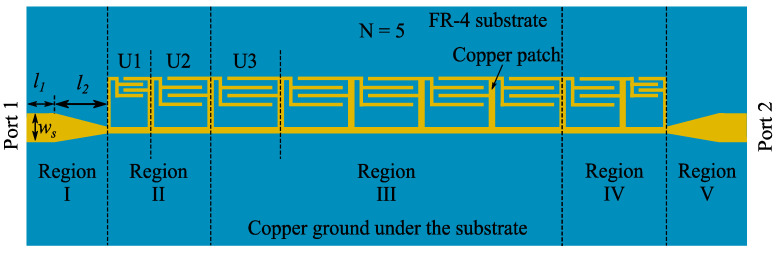
Geometry of proposed filter.

**Figure 5 micromachines-14-01687-f005:**
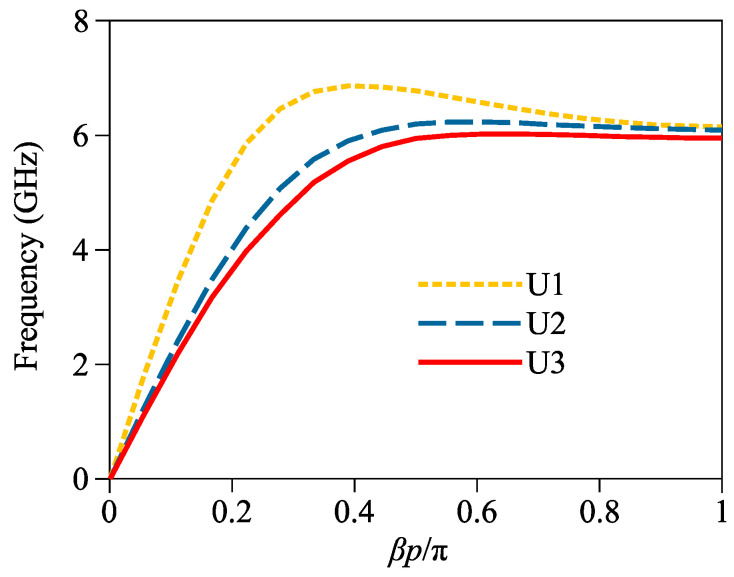
Simulated dispersion curves of U1, U2 and U3.

**Figure 6 micromachines-14-01687-f006:**
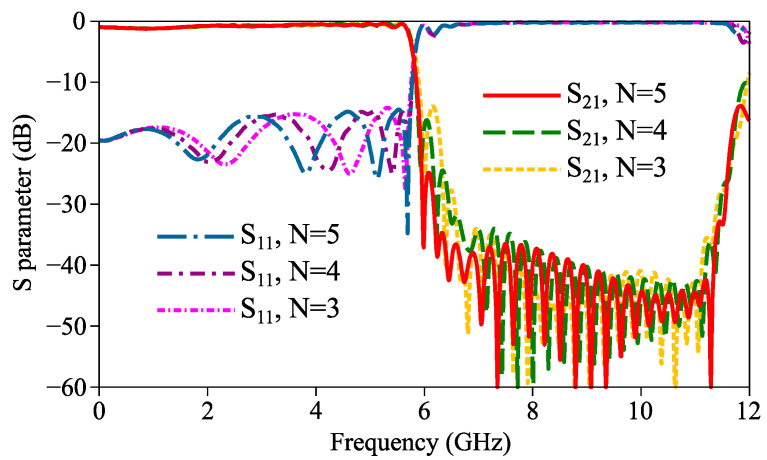
Simulated curves of S parameters of proposed filter with different N.

**Figure 7 micromachines-14-01687-f007:**
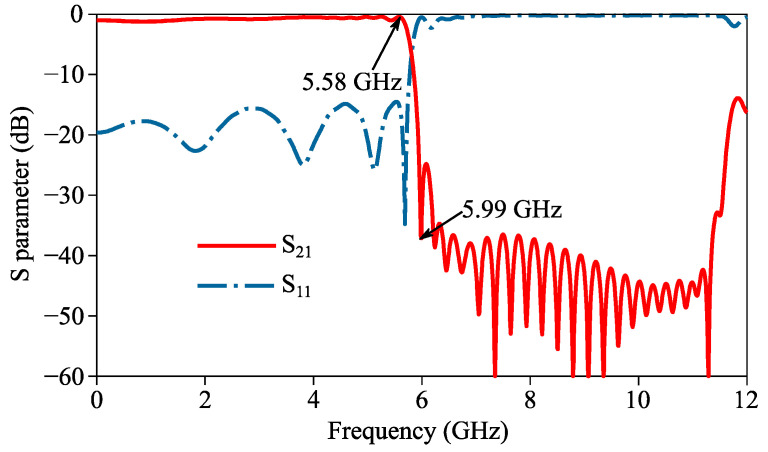
Simulated curves of S parameters of proposed filter.

**Figure 8 micromachines-14-01687-f008:**
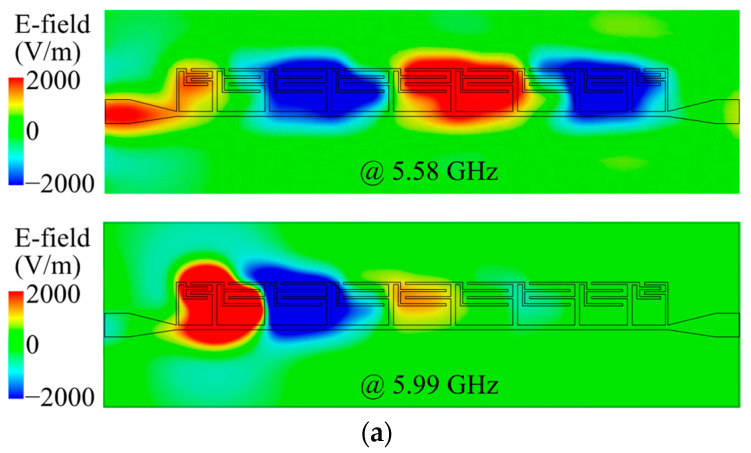
Simulated E-field and surface current distributions of proposed filter at 5.58 and 5.99 GHz: (**a**) magnitude E-field above the patch; (**b**) vector E-field above the patch; (**c**) surface current on the patch; (**d**) surface current on the ground.

**Figure 9 micromachines-14-01687-f009:**
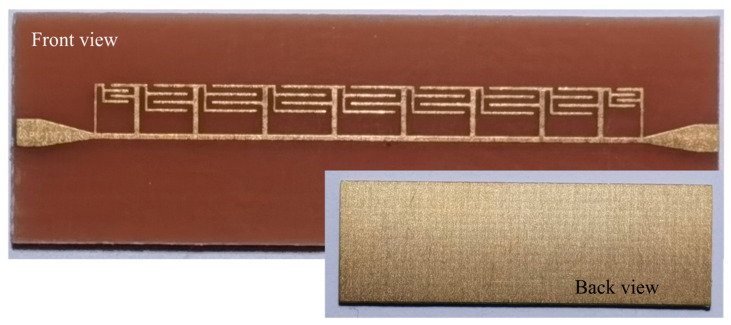
Photos of fabricated prototype of proposed filter.

**Figure 10 micromachines-14-01687-f010:**
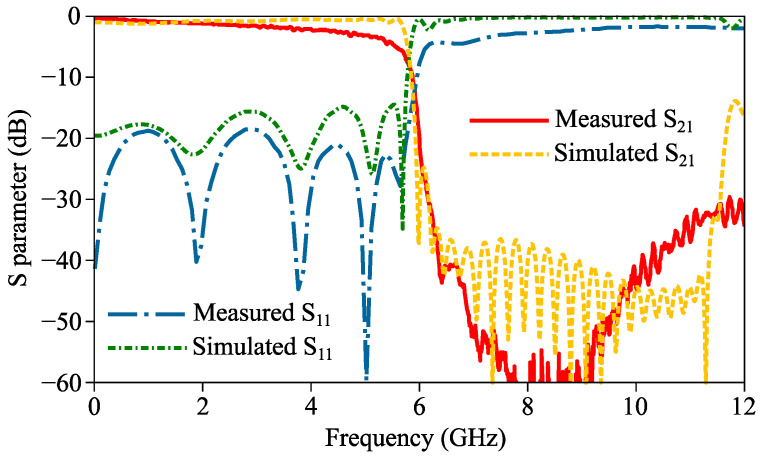
Measured curves of S parameters of proposed filter.

**Table 1 micromachines-14-01687-t001:** Dimensional parameters of U1, U2 and U3.

Unit	*p* (mm)	*h* (mm)	*d* (mm)	*g*_2_ (mm)	*w*_2_ (mm)
U1	1.6	2	0.1	0.1	0.1
U2	2.3	2	0.1	0.2	0.1
U3	2.5	2	0.1	0.2	0.1

**Table 2 micromachines-14-01687-t002:** Comparison with other band-pass and low-pass SSPP filters operating in microwave bands.

Ref.	Passband (GHz)	Return Loss in Passband (dB)	Out-of-Band Suppression (dB)	2D Size (λ_0_^2^)
[17]	0~4.6	>10	>40	3.18 × 0.38
[18]	0~8.9	>14.5	>25	3.12 × 0.34
[19]	6~14	>10	Not given	1.87 × 0.47
[20]	0.25~4.5	>12	>25	1.31 × 0.19
[23]	0~4.84	>15	>25	1.51 × 0.21
This work	0~5.74 (sim.)	>14 (sim.)	>24 (sim.)	0.5 × 0.16
	0~5.53 (mea.)	>20 (mea.)	>35 (mea.)	

## Data Availability

All data are included within the manuscript.

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
