# Peer review of "Ultra-Compact Low-Pass Spoof Surface Plasmon Polariton Filter Based on Interdigital Structure"

_micromachines, 2023, doi:10.3390/mi14091687_

Round 1

Reviewer 1 Report

In this manuscript (ID: 2538963), the authors propose and experimentally demonstrate an ultra-compact low-pass spoof surface plasmon polariton (SSPP) filter based on interdigital structure (IS). Their measurement results verify well their proposed scheme, that is, a cut-off frequency of 5.53 GHz achieved by an ultra-compact geometry of 0.5 × 0.16 square wavelength. Overall, the results of this manuscript are technically sound, and the topic of functional SSPP device could be of interest of the readers in microwave and terahertz communities. I recommend the publication of this manuscript and here are some suggestions for the authors:

1.      It will be nice if the authors can provide more detailed analysis of the working mechanism of IS. Which is the limitation of the geometric size?

2.      I noticed that there is a whole metal ground of the design, which is different from previous SSPP works. Could the authors provide more simulated E-field distributions to confirm that such a design can still support the propagation of SSPP mode.

3.      There are some recent papers about SSPPs, which are suggested to be cited so as to set a more enriched reference background: Photonics Insights 2, R04 (2023); Results in Physics 43, 106044 (2022); Advanced Optical Materials 10, 2102561 (2022).

Author Response

The authors would like to thank the Editors and Reviewers for their constructive feedback on our article. We tried our best to address all comments to their satisfaction in the revised manuscript.

Replies and Actions for the Comments of Reviewer 1

(1) It will be nice if the authors can provide more detailed analysis of the working mechanism of IS. Which is the limitation of the geometric size?

Reply: The authors want to thank the reviewer for the valuable suggestion. First of all, as described in Section 2, using IS can directly reduce the asymptotic frequency of the IS-based SSPP unit as it is of a larger capacitance between its two interdigital arms. And as shown in Figure 3, by increasing p or decreasing g2, the asymptotic frequency can be further decreased. However, decreasing g2 requires a higher fabrication accuracy which can lead to the rise of the production costs, and increasing p can lead to the increase of the filter length which goes against the original intention of the work.

Secondly, the unit number N in Region III is also optimized through simulations as shown in Figure 6 to balance the filter performance in terms of the out-of-band suppression and overall filter length.

To conclude, p, g2 and N are the main determining parameters for the proposed filter, and their values need to be chosen based on a full consideration over the balance between (a) the obtained filter performance in terms of cut-off frequency and out-of-band suppression and (b) the practical fabrication accuracy and filter length.

Action: Several sentences have been added in the last paragraph of Section 2: “Ideally, the cut-off frequency for the IS-based SSPP unit can be further reduced by in-creasing p or decreasing g2. However, a lower g2 requires higher fabrication accuracy with higher production cost, and increasing p can lead to an increase of the filter length which is obviously not wanted in this work. Therefore, their values need to be chosen considering the balance between the cut-off frequency and practical fabrication accuracy and overall length of the designed filter.”

Several sentences have been in the last paragraph of Section 3 have been adjusted: “On the other hand, increasing N directly lead to an increase of the filter length. There-fore, N is determined to be 5 to achieve both a decent out-of-band suppression over 24 dB and a relatively small filter length.”

A sentence has been added in Section 6: “Simulations show that p, g2 and N are the determining parameters for the proposed filter, and their values are optimized with a full consideration over the balance be-tween filter performance, fabrication accuracy and filter length.”

(2) I noticed that there is a whole metal ground of the design, which is different from previous SSPP works. Could the authors provide more simulated E-field distributions to confirm that such a design can still support the propagation of SSPP mode.

Reply: We want to thank the reviewer for the comments. In Section 1 and Section 6, several previous low-pass and band-pass SSPP filters [17-20, 23] have been introduced and compared with this work, and they were all designed with full metal ground. And the present Figure 8 (a), (b) show the propagation of the SSPP mode at 5.58 GHz.

Action: Simulations on the E-field and surface current have been carried out as shown in the revised Figure 8.

(3) There are some recent papers about SSPPs, which are suggested to be cited so as to set a more enriched reference background: Photonics Insights 2, R04 (2023); Results in Physics 43, 106044 (2022); Advanced Optical Materials 10, 2102561 (2022).

Reply: The authors want thank the reviewer for the suggestion. The second and third suggested paper are quite suitable for our manuscript. The first suggested paper, however, seems not related to the present topic. Could it be a clerical error?

Action: Two suggested papers are added into the reference list as reference [7], [8] and cited in the first paragraph of Section 1.

Reviewer 2 Report

The manuscript “Ultra-Compact Low-Pass Spoof Surface Plasmon Polariton Filter Based on Interdigital Structure” by Zhou-Hao Gao et al investigates an ultra-compact low-pass spoof surface plasmon polariton filter is of the small dimensional parameters for its integration in the radiofrequency circuits. The authors propose experimental prototype of filter and demonstrate the comparison of S-parameter simulated via CST Microwave Studios with real experimental measurements.

The topic of the article is relevant and the strength of the article is its engineering part. The difference between simulated S-parameter and measurement is explained as limited of fabrication, which is quite reasonable. The authors show the comparison with other filters operating in microwave bands and focus on size as an advantage.­

The references are quite recent, but could be more extensive. In particular, one could pay attention to plasmon resonance in the terms of scaling, since it appears in the title of the article (see, for example, https://doi.org/10.1364/OE.24.007133 (Lagarkov et al, Optics Express, V. 24 (7, pp.7133-7150 (2016) for a proposed microwave "plasmonic" metamaterial and comparison with real plasmonic material or similar work).

The authors correctly says that in the microwave the metal is opaque and acts as perfect electrical conductor instead of metals in optical range. However, it would be useful to talk about how a similar reduced filter will behave in the optical range. It is well known that copper has a significant imaginary part of the permeability in the optical range and is not such a good plasmonic material. In the case of scaling, what material do the authors recommend to realize a such effect in optics and what result do they expect?

Some minor remarks:­

-       Figures 6,10 could be clearer, dashed lines for different N merge into each other;

-       The authors could show the direction of the electric field in Fig. 8 for better readability.

Overall, good experimental work was done, the manuscript can be considered for publishing in Micromachines, and I support publication with minor edits.

Author Response

The authors would like to thank the Editors and Reviewers for their constructive feedback on our article. We tried our best to address all comments to their satisfaction in the revised manuscript.

Replies and Actions for the Comments of Reviewer 1

(1) The references are quite recent, but could be more extensive. In particular, one could pay attention to plasmon resonance in the terms of scaling, since it appears in the title of the article (see, for example, https://doi.org/10.1364/OE.24.007133 (Lagarkov et al, Optics Express, V. 24 (7, pp.7133-7150 (2016) for a proposed microwave "plasmonic" metamaterial and comparison with real plasmonic material or similar work).

Reply: We want to thank the reviewer for the valuable suggestion. Our article aims to provide a cheap and simple miniaturization solution of the SSPP filter for its integration into large-scale circuits for new generation wireless communication networks. And the main work of our article is to optimized the geometry of the IS-based SSPP units for the filter. This is why we selected those SSPP filters which can be fabricated by mature PCB processes for comparison.

On the other hand, the article mentioned by the reviewer introduced a dielectric SERS metamaterial consisting of periodic dielectric bars deposited on the metal substrate with interesting plasmon resonance characteristics in the microwave band. This article is very inspiring for our future research work, but is not suitable for detailed comparison in this article as its requires complicated fabrication processes.

Action: Not needed.

(2) The authors correctly say that in the microwave the metal is opaque and acts as perfect electrical conductor instead of metals in optical range. However, it would be useful to talk about how a similar reduced filter will behave in the optical range. It is well known that copper has a significant imaginary part of the permeability in the optical range and is not such a good plasmonic material. In the case of scaling, what material do the authors recommend to realize a such effect in optics and what result do they expect?

Reply: We thank the reviewer for the comments. According to the existing references, the SPP modes can be directly excited and travel along the interface between two materials with opposite permittivities (such as metal-dielectric interfaces). Based on our unprofessional understanding in the field of physics and materials, gold, silver and graphene are common metal materials used for SPP optical devices in the existing literature. Our article mainly discusses the realization of the SSPP mode with geometry optimization on the copper patch, and copper acts as PEC in the investigated microwave range. Therefore, the corresponding detailed discussions about the material are not included in our article.

Action: Not needed.

(3) Figures 6,10 could be clearer, dashed lines for different N merge into each other;

Reply: We thank the reviewer for the suggestion. However, all simulated |S11| curves basically overlap with each other in the low-frequency passband, and it is barely possible to show the difference between these curves. And the |S21| curves in the stopband also show that same problem. Therefore, all we can do is to show all curves in different colors.

Action: The curve colors in Figure 6 and Figure 10 have been adjusted for better readability.

(4) The authors could show the direction of the electric field in Fig. 8 for better readability.

Reply: We thank the reviewer for the suggestion.

Action: Simulated vector E-field distributions have been added in Figure 8 as a subfigure.
